# Impact of SARS-CoV-2 Pandemic on the Management of Patients with Hepatocellular Carcinoma

**DOI:** 10.3390/jcm11154475

**Published:** 2022-07-31

**Authors:** Maria Guarino, Valentina Cossiga, Mario Capasso, Chiara Mazzarelli, Filippo Pelizzaro, Rodolfo Sacco, Francesco Paolo Russo, Alessandro Vitale, Franco Trevisani, Giuseppe Cabibbo

**Affiliations:** 1Gastroenterology and Hepatology Unit, Department of Clinical Medicine and Surgery, University of Naples “Federico II”, 80138 Naples, Italy; valentina.cossiga@gmail.com (V.C.); mario.capa05@gmail.com (M.C.); 2Hepatology and Gastroenterology Unit, ASST Grande Ospedale Metropolitano Niguarda, 20162 Milan, Italy; chiara.mazzarelli@ospedaleniguarda.it; 3Department of Surgery, Oncology and Gastroenterology, University of Padova, 35122 Padova, Italy; filippo.pelizzaro@gmail.com (F.P.); francescopaolo.russo@unipd.it (F.P.R.); 4Gastroenterology Unit, Azienda Ospedale-Università di Padova, 35128 Padova, Italy; 5Gastroenterology and Endoscopy Unit, Policlinico Riuniti, 71122 Foggia, Italy; saccorodolfo@hotmail.com; 6Hepatobiliary Surgery and Liver Transplantation Unit, Department of Surgery, Oncology and Gastroenterology, University of Padova, 35122 Padova, Italy; alessandro.vitale@unipd.it; 7Unit of Semeiotics, Liver and Alcohol-Related Diseases, IRCCS Azienda Ospedaliero-Universitaria di Bologna, 40138 Bologna, Italy; franco.trevisani@unibo.it; 8Unit of Semeiotics, Liver and Alcohol-Related Diseases, Department of Medical and Surgical Sciences, Alma Mater Studiorum-University of Bologna, 40126 Bologna, Italy; 9Section of Gastroenterology and Hepatology, Department of Health Promotion, Mother and Child Care, Internal Medicine and Medical Specialties (ProMISE)-University of Palermo, 90133 Palermo, Italy; giuseppe.cabibbo78@gmail.com

**Keywords:** hepatocellular carcinoma, COVID-19, SARS-CoV-2, liver cancer

## Abstract

Worldwide, the severe acute respiratory syndrome coronavirus 2 (SARS-CoV-2) significantly increases mortality and morbidity. The Coronavirus Disease 2019 (COVID-19) outbreak has had a considerable impact on healthcare systems all around the world, having a significant effect on planned patient activity and established care pathways, in order to meet the difficult task of the global pandemic. Patients with hepatocellular carcinoma (HCC) are considered a particularly susceptible population and conceivably at increased risk for severe COVID-19 because of two combined risk factors: chronic advanced liver disease and HCC itself. In these challenging times, it is mandatory to reshape clinical practice in a prompt way to preserve the highest standards of patient care and safety. However, due to the stay-at-home measures instituted to stop the spread of COVID-19, HCC surveillance has incurred a dramatic drop, and care for HCC patients has been rearranged by refining the algorithm for HCC treatment to the COVID-19 pandemic, permitting these patients to be safely managed by identifying those most at risk of neoplastic disease progression.

## 1. Introduction

The spread of the global pandemic by Severe Acute Respiratory Syndrome Coronavirus 2 (SARS-CoV-2) and the subsequent coronavirus disease 2019 (COVID-19) has rapidly become a public health concern with several unmet issues. Since the beginning, the European Association for the Study of the Liver (EASL), as well as several national and international Hepatology Societies, has published position papers to provide guidance for physicians involved in the management of patients with chronic liver diseases (CLD) during the COVID-19 pandemic [1]. Nonetheless, during these two years, the COVID-19 outbreak has forced hepatologists to rethink the allocation of their resources and, in the setting of hepatocellular carcinoma (HCC) management, several modifications have been adopted to harbor the crisis and minimize patients’ exposure to the infection, during and after recovery from the pandemic. The aim of this review is to furnish an overview of the risks of SARS-CoV-2 infection in HCC patients and of the impact of the pandemic on HCC patient management with regard to surveillance programs, diagnosis, and treatment.

## 2. Risks of COVID-19 and Serious Illness from COVID-19 in Patients with HCC

It is generally known that SARS-CoV-2 infects host cells binding to the cell receptor angiotensin-converting enzyme II (ACE2), widely expressed in the lung, and also in the gastrointestinal tract [2]. In the liver, ACE2 is mainly expressed by cholangiocytes, but the SARS-CoV-2 infection does not lead to bile duct injury. Indeed, histological features, found in liver biopsy, such as microvesicular liver steatosis, mild lobular, and portal activity, closely resemble a drug-induced liver injury (DILI) [3]. Therefore, it is possible to say that interaction with the ACE2 receptor (ACE2R) is the way, used by SARS-CoV-2, to infect host cells, not the way used for liver damage. Liver damage is probably established through virus direct action or an immuno-mediated mechanism and it may involve both healthy people and patients with CLD [2]. It is also known that during the COVID-19 disease there is a ‘cytokine storm’ with high levels of the proinflammatory cytokines (interleukin 1 and 6, tumor necrosis factor, etc.) whose nature is still poorly understood [4]. All these mechanisms involved in liver injuries during COVID-19 infection suggest that it is probably multifactorial damage, including vascular damage, DILI, coagulopathy, a direct cytopathic effect of the virus, as well as an exaggerated systemic immune response [4]. The main biochemical abnormality detected is an elevation in aminotransferase levels, often mild, observed in more than 20% of cases, with a subsequent minimal increase in bilirubin levels (10% of patients), while the increase in gamma-glutamyl transferase and alkaline phosphatase is uncommon [2,5]. Liver damage is not responsible for high mortality or severe illness in patients with SARS-CoV-2 infection, but it seems that COVID-19 infection may affect liver disease progression [6]. The main mechanism is linked to an exaggerated release of inflammatory cytokines and the activation of the inflammasome pathway in target cells, following SARS-CoV-2 infection, mostly in patients with cirrhosis [6]. In particular, patients with cirrhosis have high overall mortality for COVID-19 (32%), increasing with each Child–Pugh class (patients with CLD without cirrhosis, as well as general population: 8%, Child–Pugh A 19%, Child–Pugh B 35%, Child–Pugh C 51%) [7]. Furthermore, acute-on-chronic liver failure develops in up to 12–50% of COVID19 patients with CLD, with liver disease severity as a strong predictor of developing severe COVID-19 [7]. Regarding patients with HCC, Kim et al. demonstrated that HCC is an independent predictor of death in patients with COVID-19 (HR 3.31, 95%, CI 1.53–7.16) [8]. In patients with advanced CLD, the prognosis of COVID-19 is more related to the therapeutic effort and the subsequent liver disease progression, and less to real liver dysfunction [9].

In HCC patients, ACE2R is highly expressed in cancer tissue, and it could make such patients more vulnerable to SARS-CoV-2 infection, even if ACE2R is not the only pathway of infection. More recently, NRP1 (neuropilin-1) has been detected as a facilitating receptor for SARS-CoV-2 infection. Both ACE2R and NRP1 may be overexpressed in liver cancer cells (particularly by nonparenchymal cells, such as Kupffer’s cells, hepatic stellate cells, and liver sinusoidal endothelial cells) and it may increase the risk of SARS-CoV-2 infection in HCC patients. Despite that, in the literature, there was no strong evidence of an increased risk of infection in HCC patients [10]. In their study, Fründt et al. [11] analyzed the incidence of COVID-19 in patients (outpatients or inpatients) with liver cirrhosis (with or without HCC) demonstrating that the incidence was lower than expected, and that routine medical care does not increase the incidence of infection, thanks to preventive measures. Nonetheless, this study is limited by the single-center design and by the small sample size [11].

Therefore, even if a higher incidence of SARS-CoV-2 infection in patients with HCC is doubtful, several studies have demonstrated a worse outcome (in terms of mortality and serious illness) in patients with CLD and in those with HCC [12]. In particular, patients with HCC infected by SARS-CoV-2 have an increased risk of complications, intensive care unit (ICU) admission, and lethal outcomes compared to patients without cancer [10]. In a French retrospective study, enrolling more than 15,000 patients with CLD (of them, 3693 with alcohol-use disorders), Mallet et al. showed that the 30-day mortality after COVID-19 is significantly increased in patients with alcohol use disorders, alcohol-related liver disease, cirrhosis, and HCC [9]. The higher overall mortality in patients with CLD was demonstrated also by a multicentric study conducted in the United States showing that alcohol-related liver disease, decompensated cirrhosis, and HCC were independent risk factors of higher overall mortality with a sevenfold increased risk of death from COVID-19 in HCC patients compared to patients with different CLD [13]. Recently, Iavarone et al. showed higher 30-day mortality rates in cirrhotic patients with COVID-19 than in those with bacterial infections (34% vs. 17%), as well as when comparing cirrhotic COVID-19 positive patients vs. non cirrhotic ones (34% vs. 18%), especially for those with pulmonary failure and with worsening liver function at COVID-19 diagnosis [14]. Such a higher risk of serious illness increases also considering the cohort of patients with ongoing liver damage as demonstrated by Guler et al. [15], observing that patients with HCC and active HCV infection show up with severe COVID-19 (such as pneumonia and severe pulmonary disease). Recently, a multicenter international retrospective study including the largest cohort of patients with liver cancer and COVID-19 infection (250 patients before vaccination) described data about mortality rates of HCC patients undergoing different oncological treatments during the first wave of the pandemic (February to December 2020) [16]. The 30-day mortality rate was 12.96% in de-novo HCC patients (those who received a diagnosis of HCC coincidentally with SARS-CoV-2 infection) and 20.25% in those with HCC history. The 30-day mortality rate showed a stepwise trend along Child–Pugh classes and along the Barcelona Clinic Liver Cancer (BCLC) stages. Additionally, 33.7% of patients with previous HCC developed HCC recurrence during the follow-up, while 18.4% of patients with HCC (history or de novo) died within the first 30 days of COVID-19 diagnosis [16].

In conclusion, the incidence of SARS-CoV-2 infection does not show a real increase in patients with CLD or in HCC patients. However, COVID-19 in HCC cirrhotic patients may have a worse prognosis than in the general population in terms of severe illness, complications of SARS-CoV-2 infection, and mortality. Hence, the importance of implementing measures to reduce the risk of infection in these patients.

## 3. Measures to Reduce the Risk of COVID-19 in HCC Patients

Because of the higher mortality risk among HCC patients, several measures to minimize the risk of SARS-CoV-2 infection have been proposed during the pandemic. All Scientific Societies give suggestions to limit COVID-19 in this population, resumable in four points (Figure 1):General measures;Use of telemedicine;Changing management, diagnostic, and therapeutic algorithms;Vaccination.

General measures for all patients, such as physical distancing, staying away from closed spaces without a face mask, handwashing, and correct education on infection preventive measures, may be enough for preventing the infection in CLD and HCC patients [2,17]. Measures required largely differ across regions and individual institutions, according to the level of effort imposed by the pandemic (for example, the availability of resources such as personal protective equipment and inpatient beds as well as the number of inpatients infected with SARS-CoV-2) [17].

Since March 2020, one of the first medical activities used for diminishing the incidence of the SARS-CoV-2 infection was telemedicine. The policy suggested by the EASL recommends the proper use of telemedicine to minimize hospital visits and avoid hospital admission [1]. That was the best choice for following up outpatients with CLD or those with a history of HCC and for screening patients who really need hospital admission, such as for decompensated cirrhosis or specific HCC treatments such as systemic treatments [18,19]. Ponziani et al. [20], as well as Aghemo et al. [21], conducted a web survey on a large number of Italian centers, showing a reduction in diagnostic and therapeutic procedures, and an increased use of telemedicine for patients’ follow-up, during both the first and the second pandemic “wave”. However, remote medical monitoring does not allow for a physical examination and telemedicine cannot replace medical performance, so it cannot be considered a definitive solution [22]. Although telemedicine has constituted a valid alternative during the COVID-19 pandemic, there are several characteristics to consider in the choice of visit type. First of all, old patients may not be able to use the devices for telemedicine and the presence of adequate family support must be investigated. Therefore, the age of the patients and the distance to the hospital are factors to be considered to choose the best management [23]. Otherwise, some patients desire teleconsultations also as part of their future care, because of reduced cost and travel time [24]. In brief, the switch from an in-person visit to a teleconsultation should be discussed with the patient considering his willingness and should be evaluated on a case-by-case basis.

According to the prevalence of community transmission of COVID-19, the HCC management programs should be rearranged. Treatment decision making for HCC should take into account the risk of infection, the availability of medical personnel, and the risk/benefit ratio for a single patient [18]. In view of preventing SARS-CoV-2 infection in HCC patients, changes in diagnostic and therapeutic algorithms have been proposed, and that was crucial in the management of HCC patients in the pandemic [25]. Often the main decision was to delay, when possible, the surgical treatment, but the American Association for the Study of Liver Diseases (AASLD) and EASL recommend continuing HCC surveillance and treatment with an acceptable delay of a maximum of two months to reduce the number of patients presenting with HCC not amenable to treatment [1,26]. The general agreement was to delay surgery, promoting locoregional treatment, such as ablation or trans-arterial options, considering two main causes: (a) the day of hospitalization, and (b) the need for post-operative ICU care, often busy for COVID-19 patients [27,28]. The delayed treatment must be considered in the non-urgent treatment of localized HCC by 2–3 months if there is no negative impact on the oncological outcomes; otherwise, large or multifocal HCC candidate for resection must have a “high surgical priority” [29]. In this context, a stratification system may be used to identify “at risk” patients for whom surveillance should be prioritized to avoid a delay in surgical or ablative treatment [29]. Barry et al. [30] proposed different treatment recommendations for every BCLC stage: for early BCLC stages (0-A), they prefer surveillance, ablation with microwaves or radiofrequency, or other bridge locoregional treatment instead of surgical or liver transplant option, while for BCLC B or C, the usual treatment options can be maintained. Iavarone et al., in a single center report, describe measures adopted to prevent SARS-CoV-2 infection in HCC patients [31]. They managed the admission for HCC treatment as follows: using protective equipment, SARS-CoV-2 tested by nasopharyngeal swabs regularly (every 1–2 weeks), well-defined COVID-19-free areas of the hospital with dedicated healthcare staff, a telephone survey aiming to detect possible contact or COVID-19 infection, and a mandatory SARS-CoV-2 test performed the day before admission for patients. Whenever possible, they preferred radiofrequency ablation to surgery, without the detrimental consequences for the radiological response of HCC patients [31]. They demonstrated that these strategies were able to safely ensure continuity of care for patients with HCC.

Finally, another measure to prevent infection is the use of vaccines against SARS-CoV-2. Vaccines demonstrated immunogenicity and efficacy in CLD patients, especially patients with advanced liver disease [32] or immunosuppressed [33]. So, since the start of the vaccination era, EASL has recommended vaccination for patients with CLD, liver transplant recipients, and hepatobiliary cancer, as a priority [34]. Although the number of patients with cancer enrolled in the vaccination trial is small, all scientific societies have recommended vaccines against SARS-CoV-2 in oncologic settings [35,36]. To confirm that, Pinato et al. recently published data from the retrospective, multicenter, OnCovid registry study about the outcomes of the SARS-CoV-2 infection among vaccinated and unvaccinated patients with cancer in Europe [37]. Compared to patients diagnosed during the pre-vaccination period, such as those described by Munoz-Martinez et al. [16], boosted and vaccinated patients had significant improvements in case-fatality rates at 2 weeks (9% vs. 23%) and 4 weeks (13% vs. 29%), complications due to COVID-19 (15% vs. 39%), and hospitalization due to COVID-19 (24% vs. 56%); moreover, they had fewer requirements for COVID-19-specific therapy (33% vs. 65%) and oxygen need (22% vs. 54%) [37]. Additionally, John et al. showed that patients with cirrhosis and post-vaccination COVID-19 infection have lower rates of complications, including hospitalization, mechanical ventilation, or death than those with unvaccinated COVID-19, even after partial vaccination [32]. These findings support the universal vaccination of patients with cancer (such as HCC) as a protective measure to reduce morbidity and mortality from COVID-19. For these reasons, all limitations in HCC management addressed by Scientific Societies in the pre-vaccination era may be abandoned, and we can look forward to a gradual return to the normal management of HCC patients, thanks to vaccinations.

In conclusion, the common attitude of Liver Units was to follow up HCC patients using remote evaluation, as promptly as possible, promoting vaccination. On the other hand, during the pandemic, it was necessary to assess a different diagnostic–therapeutic approach in order to not miss the curative aim. The main difference has been shown mostly in early BCLC stages in which locoregional treatments showed a crucial role. All these measures have been helpful in decreasing the risk of SARS-CoV-2 infection in HCC patients, avoiding hospitalization and ICU care related to COVID-19.

## 4. Impact of COVID-19 on HCC Surveillance and Diagnosis

The institution of measures, such as social distancing and stay-at-home orders, implemented to prevent the spread of COVID-19, have adversely affected the routine outpatient care of CLD, including the HCC surveillance, with short- and long-term effects on mortality from liver disease [38]. All society guidelines recommend semiannual surveillance for HCC [39,40], even though the overall worth of HCC screening in cirrhotic patients is still a matter of debate in light of the absence of specific randomized-controlled trials. The aim of HCC surveillance is to decrease HCC-related mortality by promoting very-early tumor detection for giving patients the possibility of curative treatments. Recently, in a systematic review of several cohort studies, Singal et al. showed that HCC semi-annual surveillance is associated with significant improvement in early diagnosis, curative treatment, and overall survival in cirrhotic patients [41]. During the COVID-19 pandemic, most health systems postponed elective imaging, including HCC screening. The COVID-19 outbreak has largely limited the medical care of these patients, with consequences ranging from early diagnosis to treatment. As ultrasound surveillance has mainly been delayed indefinitely, the risk of diagnosing HCC at an advanced stage increased in almost 25% of cirrhotic patients [42].

An international survey conducted in 76 Liver Units and focused on the impact of COVID-19 on the management of patients with HCC showed that 87% of the centers modified their clinical practice during the COVID-19 pandemic. Particularly, 80.9% of them modified their HCC screening program and 40.8% changed their diagnostic procedures [23]. Moreover, six retrospective studies analyzed the delay in the HCC diagnosis during the COVID-19 outbreak [8,43,44,45,46,47] (Table 1). All of them, except one, showed a significant reduction in HCC diagnosis during the pandemic compared to the periods before COVID-19. Particularly, Mahmud et al. [43] demonstrated that the proportion of patients completing HCC surveillance was significantly lower in 2020 vs. 2019 for each analyzed month. In multivariate analysis, increased odds of surveillance were related to age > 60 years, cirrhosis decompensation, and later 2020 month. Similarly, Ribaldone et al. [45] showed that the number of HCC diagnoses, in an Italian Tertiary Center, diminished in the periods March–December 2020 and January–October 2021 compared to May 2019–February 2020 due to a reduction in regular ultrasound surveillance. However, there were no significant changes in HCC characteristics (the BCLC stage remained the same). Finally, Kim et al. [8] showed, in a large cohort of American Veterans, a 44% decline in HCC screening, irrespective of liver disease severity. Only Kuzuu et al. [47] showed, in a Japanese cohort, a non-significant decrease in HCC diagnosis during vs. before the COVID-19 pandemic.

Currently, the AASLD and the EASL recommend continuing HCC surveillance, deferring it by 8–12 weeks during times of limited radiologic capacities such as COVID-19 [1,26]. However, given the heterogeneous growth pattern of HCC, the choice to postpone surveillance should be evaluated on a case-by-case basis [48]. Some authors suggest stratifying the risk of HCC using scores to identify patients at high risk, for whom surveillance should not be postponed [49,50]. Moreover, due to the elevated risk of COVID-19 exposure associated with the close contact between physician and patient during ultrasound examination, the Asian Pacific Association for the Study of the Liver (APASL) recommends limiting the use of this practice for surveillance, preferring CT (computed tomography) or MRI (magnetic resonance imaging) in patients at high HCC risk [51].

In conclusion, during the COVID-19 pandemic, due to the lack of resources, HCC screening was often postponed. However, all scientific societies have recommended that the schedule followed before the pandemic be maintained and to postpone surveillance only in low-risk cases identified with appropriate risk scores.

## 5. Impact of COVID-19 on HCC Management and Treatment

The COVID-19 pandemic has tremendously limited the clinical management of HCC patients, with impacts from early diagnosis to treatment. HCC management is intrinsically and particularly complex so that the treatment is ordinarily established by a multidisciplinary tumor board. In addition, during pandemic waves, it becomes mandatory to balance the risk of a deferred tumor diagnosis, precluding potentially curative treatments against the infection risk. Moreover, it must be taken into account that resources usually allocated to cancer care are limited, as a consequence of their redistribution to face the pandemic.

Elective hospital admissions were postponed or canceled during the pandemic peak due to low hospital capacity and the conversion of hepatological/gastroenterological units into COVID-19 wards (Table 2). For example, only a limited number of Italian hepatologists (20%) reported no relevant changes in their daily activity during the first wave. Indeed, more than 50% of hepatological/gastroenterological units were converted into COVID-19 wards, or the number of beds was reduced, due to nurses and medical personnel shortages either for medical leave or reallocation [20,21]. A similar scenario was reported in other European countries where COVID-19 infection had a high incidence [52,53].

A significant reduction in daily activity was also reported for outpatient clinics (Table 2). In different countries, outpatient care and diagnostic procedures were unavailable for a certain time. In addition, a substantial number of patients were fearful of reaching medical facilities for the risk of getting infected. An international survey has reported that almost 90% of centers have modified their clinical practice with a significant reduction in diagnostic procedures and screening programs [23]. For this reason, a greater number of centers decided to switch to remote contact (by e-mail or phone) to facilitate the management of patients with CLD or cancer [21,23]. Despite the limited number of consultations available and the use of telemedicine, patient satisfaction and assistance perception remained high during the pandemic [19,54]. Telehealth is in fact associated with considerable patient-centered benefits, including reduced harm, decreased charges and costs, and reduced absenteeism for those who work [55].

Conversely, HCC treatment options have been insufficiently maintained (Table 2). Surgical resection and liver transplants (LT) have been the most affected ones due to a shortage of anesthetists and other healthcare workers [56]. Munoz-Martinez et al. reported that at least 50% of curative or palliative treatments for HCC were canceled during the pandemic [23]. Similar experiences are reported in surveys collected across the world. In Italy, both surgical and non-surgical loco-regional treatments have been decreased (by 44% and 34%, respectively) or suspended (by 44% and 8%, respectively) during the first and second waves of the pandemic [20,52]. In this regard, a considerable proportion of institutions, over the world, reported delays > 2 months in the treatment of HCC patients and changes in treatment modality, according to BCLC C classification or compared to similar patients managed before the pandemic [28,57]. In a shortage of hospital resources, as experienced during the pandemic, the main hepatologists’ worry was to prevent any loss of potentially curative treatment for the patient.

Consequently, clinical practice guidelines have been modified by different international societies, such as the EASL [1], the APASL [51], and the AASLD [26], adapting the therapeutic algorithm of BCLC to the new emergency situation in an attempt to select the best HCC patient for the best treatment (Table 3).

All the scientific societies recommend ensuring continuity of care for HCC patients, avoiding treatment delays or interruptions, and maintaining multidisciplinary (MD) tumor boards (Figure 2). In fact, because of the complexity of HCC management, an MD approach is recommended to optimize the care of patients with HCC. It has been well-recognized that an MD approach increases survival as compared with non-MD care [62]. Unfortunately, in many parts of the world, MD boards have been discontinued, considering that many hepatological wards were converted to COVID-19 wards or considering the need for social distancing [28,57]. However, social distancing should not be considered a motif to temporarily abolish MD meetings, as they can be carried on via the web or in person on the basis of the local pandemic situation. Deviations from ‘‘standard’’ treatment should be discussed and validated in an MD tumor meeting. The alternative therapeutic decision, as well as the benefits of the choice and their associated risks, should be discussed with the patient and accurately documented for legal purposes [31].

Reshaping HCC management on the basis of available therapeutic resources and the local current pandemic situation has been demonstrated to maintain a high standard of performance, similar to the pre-pandemic period, and to avoid futile delays in the diagnosis and treatment of cancer patients [31]. In this changed scenario, bridge therapy played a crucial role. In the case of early-stage HCCs, the capacity of most hospitals for surgical treatment (resection or LT) was reduced because of the limited number of anesthesiologists as well as shortages in ICU beds. Therefore, hepatologists have to carefully select patients who should be prioritized for surgery according to their risk of surgical complications and the need for ICU admission. Comorbidities, advanced age (75 years or greater), underlying cirrhosis, extensive blood transfusions, and the surgical act planned (minor or major hepatectomy, liver segments/sectors concerned/biliary reconstruction) are very well-known risk factors associated with surgical complications and with the risk of ICU admission after surgery [63,64,65]. A laparoscopic approach should be preferred because of its advantages in pulmonary function and length of stay in the hospital [66]. When surgery is not feasible or in patients with a high risk of complications, different approaches can be adopted. A delaying strategy relies on strict imaging monitoring in selected cases. A general agreement to postpone the non-urgent treatment of localized HCC by 2–3 months if oncological outcomes are not likely to be affected has been reached [67]. The median HCC doubling time was 229 days in a series of 242 HCC patients, with indolent growth mainly observed in large tumors with alpha-fetoprotein levels < 20 ng/mL and in patients with non-viral liver disease [68]. On the other hand, hepatologists should keep in mind that a retrospective study by Singal et al. demonstrated that a treatment delay of >3 months was associated with worse overall survival [69].

In cases where the HCC treatment cannot be postponed, locoregional approaches, such as radiofrequency/microwave ablation and trans-arterial therapies (Transarterial chemoembolization—TACE or Transarterial radioembolization—TARE), can be used as alternative or bridging methods, until surgery can be performed. In fact, in many centers locoregional treatments did not decrease but sometimes increased, such as in Pisa [27] where TACE and TARE were alternative options for patients with delayed treatment. In patients undergoing surgery or locoregional therapy, strict monitoring should be carried out with imaging examinations every 12–16 weeks in the first year and then every 6 months to evaluate treatment response and detect possible HCC recurrence. Delay in the follow-up monitoring >90 days has been demonstrated to be associated with a lower objective response rate in HCC patients [70]. In patients with a complete response to treatment and without recurrence for >2 years, imaging tests can be postponed for up to 2 months according to AASLD recommendations [26,71].

The prognosis of HCC patients mainly depends on both tumor and underlying liver disease stages. For this reason, risk stratification has a crucial role in defining the most appropriate treatment for each patient. In addition, COVID-19 infection manifesting during the post-surgical period is associated with poor outcomes. Notably, a large multicenter study enrolling 1128 COVID-19-positive patients undergoing surgery at 235 hospitals in 24 countries, showed that the overall 30-day mortality rate was 23.8%, and the rate of pulmonary complications was 51.2% [72].

During the pandemic, the number of LT was reduced due to the limited availability of anesthesiologists as well as a shortage of donors [28,73,74,75]. The United Network for Organ Sharing described a significant reduction in LT (both living and deceased donor liver) as well as an increase in patient delisting because of COVID-19-related issues and a noticeable reduction in the recovery of deceased-donor organs [76]. The major guidelines suggest a temporary suspension of elective living donor transplants aiming to protect both the donor and recipient. According to the EASL position paper, living-donor transplants should be defined on a case-by-case evaluation [1]. It is recommended that donors and recipients be both tested for SARS-CoV-2 before LT [1,51]. Considering the high risk of mortality in patients with cirrhosis [77], it is suggested that patients on a waiting list should be vaccinated for COVID-19 [34]. According to current recommendations, LT should not be postponed for high-priority HCC patients with a poor prognosis in the short term, such as those with either acute or chronic liver failure, a high MELD score, and HCC above Milan criteria [1,26]. In patients on waiting lists, when a complete response to bridging therapy is documented, LT may be postponed [78]. Cillo et al. demonstrated that, in a time of donor shortage such as the COVID-19 pandemic, the highest survival benefit is obtained in patients within Milano criteria (single HCC 2–5 cm, or 2–3 HCCs each ≤3 cm [79]. In these patients, transplants should not be postponed. Bridging therapy, such as ablation, trans-arterial embolization (TACE or TARE), and systemic therapy, can also be proposed in patients in whom delaying transplantation is necessary [80].

In the case of intermediate-stage HCC, TACE still represents the mainstay of treatment. Nonetheless, TACE is a palliative treatment to reach cytoreduction, with the main purpose of controlling the tumor growth for as long as possible. Short-term use of corticosteroids (3 days) should be prescribed in TACE patients except for those with contraindication (e.g., uncontrolled diabetes) in order to reduce post “embolization syndrome” and minimize hospital stay [81]. In the elderly (>80 years) and in patients with comorbidities, TACE/TARE might be postponed, minimizing the risk connected to hospitalization, considering the oncological benefit in comparison to the risk of COVID-19 exposure [28].

Finally, in the case of advanced-stage BCLC C, HCC patients who receive oral multitargeted tyrosine kinase inhibitors (TKIs) should continue therapy without any interruptions. It is recommended to minimize the access to the hospital for control visits and increase the use of telephone-based consultations [78]. In addition, home blood sampling and drug delivery were implemented in many centers to facilitate the management of patients on systemic therapy. Management of adverse drug reactions, such as dermato-toxicities, diarrhea, and hypertension, may be handled by community doctors after coordination with oncological teams. Despite these recommendations, some centers reported a reduction in the prescription of systemic therapy, at least during the first wave of the pandemic [21]. EASL recommendations suggest suspending temporarily immunotherapy or switching patients from intravenous drugs to orally administered formulations to avoid exposure to COVID-19 at the infusion center and to decrease the number of hospital visits [1]. In this context, the driven role of hepatologists in managing the various steps of an HCC patient’s journey is crucial [82]. Enrolment in clinical trials has been stopped during the pandemic, and these patients are recommended to be treated with TKIs, if eligible. For patients enrolled in clinical trials, rearrangement of hospital visits, treatment schedules, and routine screening measures can be considered [20,83].

In conclusion, HCC patients should be managed by a multidisciplinary tumor board, balancing the risk of deferred tumor curative treatments against the infection risk. Nonetheless, most of the data showed that HCC patients can be fittingly and timely managed even during a pandemic by implementing strategies for tailoring the most cost-effective solution on a case-by-case basis and reducing the risk of COVID-19 infection.

## 6. Conclusions

In conclusion, this review describes the prevalence and outcomes of COVID-19 in HCC patients, and the impact of the SARS-CoV-2 pandemic on liver cancer management for difficulties in prompt HCC detection and its expected subsequent increase in HCC incidence due to delayed surveillance protocols. This review also gives a comprehensive overview of adjustments to medical care for HCC patients proposed to minimize SARS-CoV-2 exposure and redirect resources to handle the outbreak. The indirect inevitable effects of the COVID-19 pandemic in the coming years have yet to be ascertained and further studies are necessary to elucidate the real impact on the prognosis and survival of HCC patients “unconventionally” managed during the last 2 years. Indeed, several questions regarding the impact of the COVID-19 pandemic on HCC remain to be answered. The most important one is: How and how much will the modifications in clinical practice imposed by this pandemic affect outcomes of HCC patients?

## Figures and Tables

**Figure 1 jcm-11-04475-f001:**
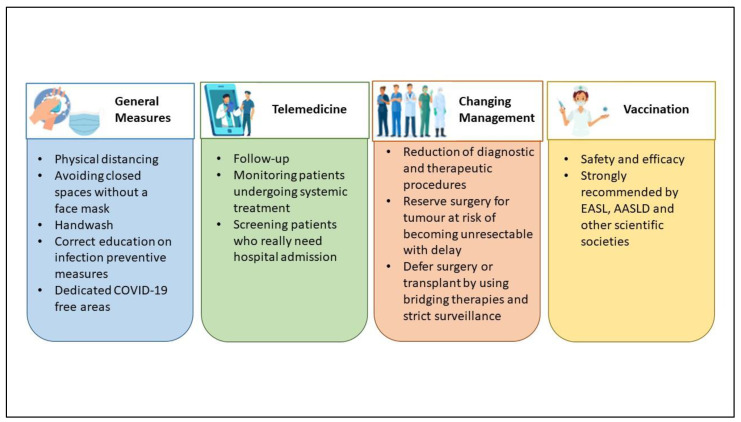
Measures to minimize the risk of COVID-19 in HCC (hepatocellular carcinoma) patients.

**Figure 2 jcm-11-04475-f002:**
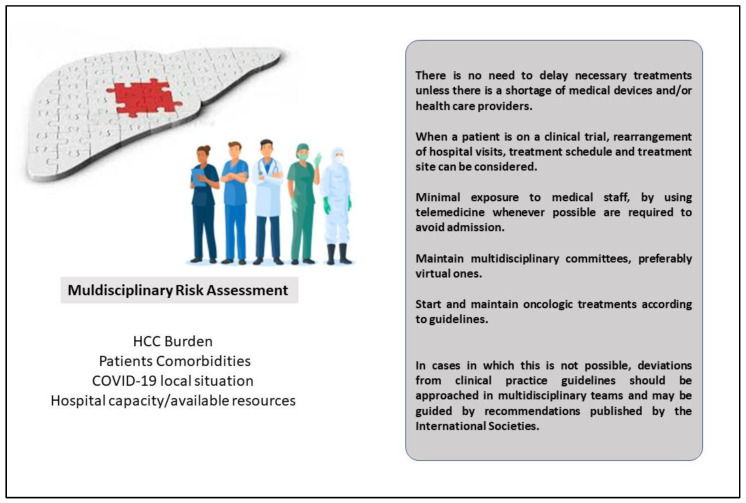
The role of Multidisciplinary Tumor Board during the COVID-19 pandemic for HCC patients.

**Table 1 jcm-11-04475-t001:** Studies evaluating the impact of COVID-19 on the HCC (hepatocellular carcinoma) surveillance and diagnosis.

Author, Year	Study Design	Number of Patients	Conclusions
Mahmud N, 2021 [43]	Retrospective	15,480	35.3% of patients completed surveillance
Toyoda H, 2020 [44]	Retrospective	14,403	39% decrease in surveillance use
Ribaldone DG, 2022 [45]	Retrospective	247	35% of patients completed surveillance
Perisetti A, 2021 [46]	Retrospective	18,818 (Pre-COVID-19), 4383 (Post-COVID-19)	Increased diagnosis of HCC during post-COVID-19 period (OR: 1.19)
Kim NJ, 2022 [8]	Retrospective	94,612 (Pre-COVID-19), 88,073 (Post-COVID-19)	44% decrease in surveillance use
Kuzuu K, 2021 [47]	Retrospective	4218 (pre-COVID-19), 949 (Post-COVID-19)	No decrease in HCC diagnosis during COVID-19

**Table 2 jcm-11-04475-t002:** Studies evaluating the impact of COVID-19 on HCC management and treatment.

Author, Year	Study Design	Number of Patients/Centers	Conclusions
Aghemo et al. (2020) [21]	Prospective web-based survey	194 Italian centers	Surgical and non-surgical loco-regional treatment procedures have been decreased (44% and 34%) or suspended (44% and 8%).
Amaddeo et al. (2021) [57]	Multicenter, retrospective, cross-sectional study	670 patients in 6 Centers in Paris	Reduction in newly diagnosed HCC (hepatocellular carcinoma) and in MD (multidisciplinary) discussion. Modification of HCC management in 13.5% of cases. Treatment delay > 30 days in 2020 compared to pre-pandemic.
Balakrishnan et al. (2020) [52]	Online survey	130 centers across Europe and Africa	Insufficient critical care capacity and reduced surgical sessions in COVID-high countries (>100,000 cases) compared to COVID-low countries.
Bargellini et al. (2021) [27]	Retrospective	Single Italian Center	27.5% reduction in MD discussion. Number of ablations was stable. 28.3% reduction in TACE. Increase use of TARE.
Crespo et al. (2020) [53]	Nationwide survey	81 Spanish centers	Outpatient visits, liver ultrasounds, and endoscopies were reduced by 81.8–91.9%. 75% decrease in therapeutic endoscopies and 89% decrease in HCC surgery, with cancelation of elective LT.
Gandhi et al. (2021) [58]	Online survey	27 centers in south-East Asia.	diagnostic delay (48.2% in BCLC 0/A/B and 51.9% in BCLC C), treatment delay (66.7% in BCLC 0/A/B and 63.0% in BCLC C), treatment modality changes (33.3% in BCLC 0/A/B and 18.5% in BCLC C). Increase of 18.3% in video/telephonic consultations.
Pomey et al. (2021) [54]	Retrospective	126 patients in a single Austrian Center	Stable number in newly diagnosed HCC. Significant delays in in person consultation and imaging screening. Reduction in MD discussion.
Zhao et al. (2021) [59]	Nationwide multicenter survey	37 centers in China	60% reduction in surgical and not-surgical activities. Significant increase in telemedicine.
Maida et al. (2020) [60]	Web-based national survey	121 Italian centers	85.1% of out-patient consultations, 96.2% of endoscopic procedures, and 72.2% of ultrasounds were limited to urgencies and oncology indications.
Munoz-Martinez et al. (2021) [23]	Web-based survey	76 centers across Europe, America, Asia, and Africa	87% of the centers modified their clinical practice, 80.9% reduced screening programs, 40.8% reduced diagnostic procedures, 50% canceled curative and/or palliative HCC treatments, and 41.7% modified the LT program. Increase in teleconsultation (51%)
Nevermann et al. (2020) [61]	Web-based survey	79 European surgical centers	60% reduction in the surgical activity compared to the pre-pandemic period. 33% reported discontinuation in MD consultation. 50% of patients reported a delay > 30 days in outpatient consultation/imaging.
Ponziani et al. (2021) [20]	Web-based survey	43 Italian centers	Locoregional or surgical HCC treatments reduced or stopped in 55.8% and 48.1% of centers, respectively.

**Table 3 jcm-11-04475-t003:** Proposed treatment recommendations by international societies according to the BCLC (Bercelona Clinic Liver Cancer) stage.

	Standard of Care According to BCLC	Proposed Treatment Recommendations by International Societies
**BCLC 0** **or** **BCLC A**	Liver Transplant Surgical resection Ablation	If an LT or surgical resection is not available, consider alternative or bridging therapy as ablation, trans-arterial embolization (TARE or TACE). In limited disease, consider surveillance
**BCLC B**	Liver transplant Locoregional therapies	Consider locoregional therapy on-demand, radiotherapy, or surveillance If trans-arterial treatments are not available, consider systemic therapy
**BCLC C**	If the patient has macrovascular thrombosis and no extra-hepatic disease, use TARE or systemic therapy; if the patient has extrahepatic disease, consider systemic therapy.	Consider systemic therapy (prescribe oral TKIs instead of immunotherapy to reduce number of visits/consultations)
**BCLC D**	Best supportive care	Best supportive care and palliative radiotherapy (in a single 8-gray fraction) for symptomatic disease

## Data Availability

Not applicable.

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
