# Peer review of "Impact of SARS-CoV-2 Pandemic on the Management of Patients with Hepatocellular Carcinoma"

_jcm, 2022, doi:10.3390/jcm11154475_

Round 1

Reviewer 1 Report

In this review, Guarino and co-workers shoed one more comorbidity (hepatocellular carcinoma) that has been affected by the COVID-19 pandemic. They highlighted a very interesting point, including the delayed diagnosis of HCC during the pandemic and the risk of infection for HCC patients. 

In my point of view, the review is well written and very relevant to the field. 

I have just one observation, the last discussion paragraph shows conclusions as well as the paragraph of the conclusions section. Rephrase the first for more clarity. 

Reviewer 2 Report

The Review by Guarino et al., summarized the risk of Covid infection in HCC patients and how the pandemic affected HCC management. Overall the review is well written and provides a deeper insights on various considerations and recommendations available in the literature.

The authors proposed a battery of valuable changes that could be performed in the clincal care of HCC patients. However, the review does not address or consider the patient willingness or reluctance for in-person visits or changes in the measures listed in Figure 1. A HCC patient will be more anxious about the infection and its associated risk while deciding on the in-person visits. A paragraph on how patient education on a case-by-case basis will help to remediate patient anxiousness in telemedicine or corrective treatment.

There are few studies that show how COVID infection may affect liver disease progression. A paragraph on this would improve the understanding of the readers why the management of HCC patients or high-risk population is crucial in the face of Covid pandemic.

Reviewer 3 Report

Please see the attached file for comments...
